# Effect of seed hydro-priming durations on germination and seedling growth of bitter gourd (*Momordica charantia*)

**Binod Adhikari**[1]*, **Pankaj Raj Dhital**[2], **Sambat Ranabhat**[1], **Hari Poudel**[1]

**1** Department of Agriculture, Agriculture and Forestry University, Chitwan, Nepal, **2** Department of Agricultural Extension and Rural Sociology, Agriculture and Forestry University, Chitwan, Nepal

* bnd8278@gmail.com

**Data Availability Statement:** All relevant data are within the paper and its Supporting information files.

## Abstract

The yield and quality of an annual crop are highly dependent on uniform and rapid germination of the seeds. In case of bitter gourd (*Momordica charantia)*, the germination and field emergence is always a problem as seeds have thick and hard seed coat. Pre-sowing hydro-priming is one of the most suitable, affordable, easily available, and cost-effective techniques in breaking down seed dormancy to enhance germination. Hence, a field experiment was conducted in Surkhet district of Nepal 2020 to assess the effect of different hydro-priming duration on germination and seedling growth of bitter gourd. The experiment was laid out in single factor Randomized Complete Block Design (RCBD) with four replications and seven treatments including different hydro-priming durations ($T_0$: control, $T_1$: 6 hours, $T_2$: 12 hours, $T_3$: 18 hours, $T_4$: 24 hours, $T_5$: 36 hours and $T_6$: 48 hours) of bitter gourd seeds of Palee variety, the most popular variety among the farmers. The highest water uptake and germination were found in 48 hours of seed hydro-primed seeds whereas the lowest water uptake and germination were observed on non-primed seeds. Similarly, the tallest seedling, most vigorous seedling in terms of seedling vigour index I and II was observed in 48 hours hydro-primed seeds followed by 36 hours of seed hydro-priming and shortest seedling and the least vigorous seedling in control. Thus 48 hours of seed hydro-priming was found to be effective for increasing germination and seedling growth in bitter gourd, which needs to be further investigated under large, open-field conditions with different varieties.

## Introduction

Bitter gourd, *Momordica charantia* (Cucurbitaceae) is a monoecious annual crop having epigeal germination, which is mostly grown from January to June. It is one of the most important summer vegetables in Nepal, which is cultivated in all ecological belts. The plant requires a warm and hot climate and grows well on sandy loam soil with pH 6.0 to 6.7 [1], and altitude range up to 1000 m with optimal germination and growth at the temperature range of 25–28˚C [2] and 24–27˚C [3] respectively.

**Funding:** The author(s) received no specific funding for this work.

**Competing interests:** The authors have declared that no competing interests exists.

The crop is of high nutritional and medicinal importance. Its immature fruit is a rich source of dietary fibers, minerals, and Vitamins (C and A) [4] which also acts as a blood purifier and is highly beneficial to diabetes patients [5]. Likewise, it also has anti-carcinogenic properties and can be used against multiple cancer forms as a cytostatic agent [6]. Moreover, it is also used in traditional medicine to correct disorders like hyperlipidemia, digestive disorders, menstrual problems, and several microbial infections [5].

Uniform and rapid germination is an important factor contributing to yield, quality, and ultimately the profit to the vegetable farmers [7]. Because of the thick seed coat, the seed consumes water gradually resulting in sluggish germination and hence field emergence is always a problem in bitter gourd, even with the seed having high germinability [8]. To overcome this problem, several techniques have been practiced and priming is one of them.

Seed priming is a simple, low-cost, and effective strategy for enhancing crop performance [9]. It is described as a physiological approach that involves hydration and drying of seeds to improve the pre-germinative metabolic process without radicle protrusion in water or solution of other priming agents [10–12]. Many agricultural and horticultural crops have been shown to benefit from it in terms of seed germination and seedling establishment and ultimately the productivity [13–17]. Early and uniform germination by break down of photo- and thermo-dormancy with extended germination temperature range, higher nutrient uptake, and improved water use efficiency have all been described as advantages of seed priming [18–20]. It effectively improves seed vigour and germination, which is a complicated agronomic feature influenced by various genetic and environmental factors [21, 22]. So, it has long been described as a potential way to promote crop performance [23]. Primed seeds also show a higher germination rate and better uniformity in emergence of seedlings which contribute to the regular establishment of crops and hence the yield. The fast growth of primed plants is related to better plant water status regulation [24] and an increased nutrient usage capacity [25].

Hydro-priming is an economic and eco-friendly technique [26] that is done by soaking seeds either in hot or in cold water for a certain period before sowing seeds in the field or any growing/nutrient media [27]. This facilitates water imbibition in seed and makes seed coat soft enough for enhanced easy and fast growth of seed embryo [28]. Moreover, the effective hydro-priming treatment causes metabolic pathways triggered during germination step II, which are then briefly stopped until a desiccation problem happens that helps to improve on-field plant behavior [29]. The short germination period, early emergence, and vigorous seedlings were observed when seeds were hydro-primed while experimenting series of crop species [30].

Each crop cultivars has its critical soaking duration which is lower than the safe limit [30]. So, knowledge of acceptable priming duration is crucial before priming seeds to achieve optimum impact. However, there is a lack of proper information on the exact duration of the seed hydro-priming, especially for the bitter gourd. Hence, the present study was conducted to find out the optimum hydro-priming duration for bitter gourd under the agro-climatic conditions of district Surkhet, Karnali Province, Nepal.

## Materials and methods

### Experimental site and weather condition

The experiment was carried out at Birendranagar Municipality-7, Itram, Surkhet. It lies at 28˚35'N to 81˚37'E. The soil was silt clay type where tomatoes were grown before the experiment.

The study site lies within Nepal's subtropical climate. It is distinguished by three distinct seasons: rainy monsoon (June-October), cold winter (November-February), and warm spring (March-May). Fieldwork was conducted from March 16 to April 15, 2020. The figure below

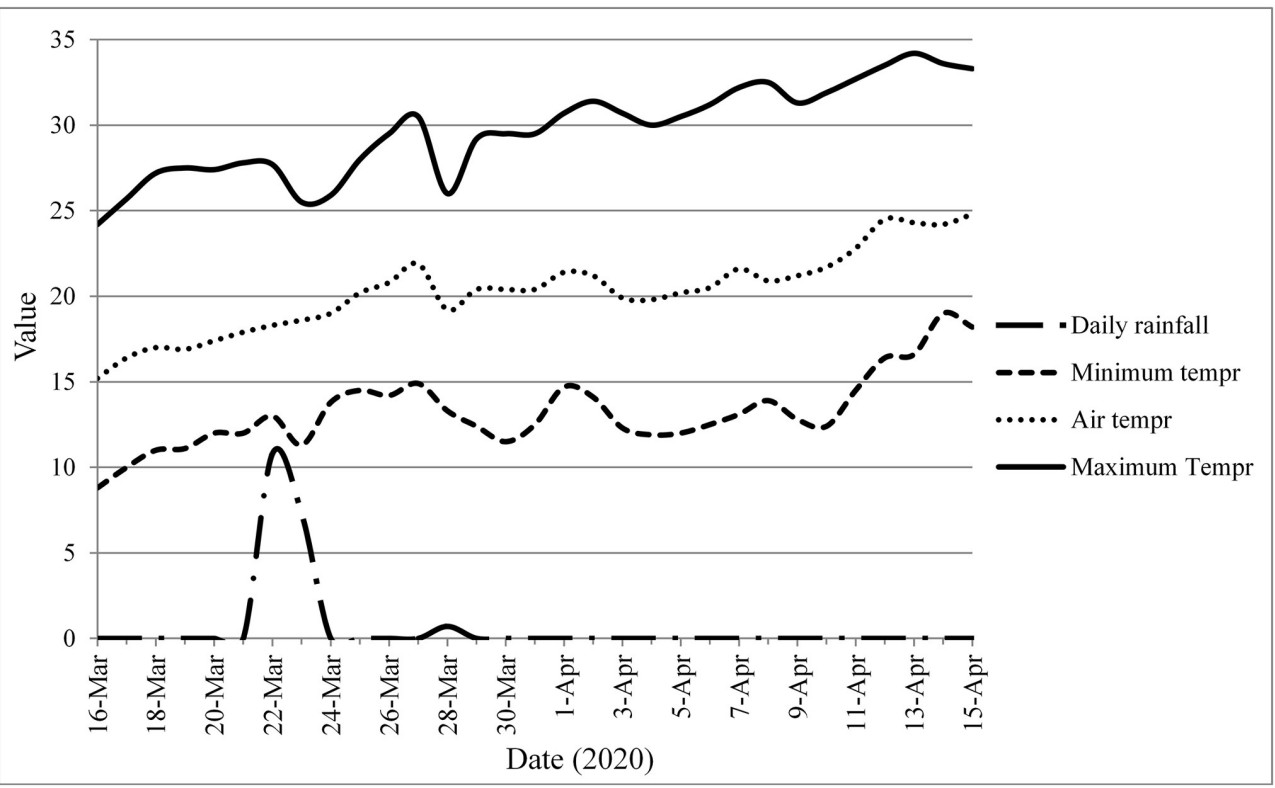

**Fig 1. Meteorological data of Birendranagar from March 16 to April 15, 2020.**

(Fig 1) shows the meteorological data during the research period which were obtained from Meteorological Forecasting Divison, 2020.

## Seed source

A bitter gourd seed of cultivar Palee F1 was bought from a local agro-veterinary shop. The seeds had a minimum of 60% of germination with at least 98% and 95% of physical and genetic purity.

## Experimental setup and recording of data

The experiment was laid out in single factor Randomized Complete Block Design (RCBD) having 7 treatments and treatments were replicated four times in open field conditions. Treatments were designed based on hydro-priming durations as hydro-priming for 6, 12, 18, 24, 36, and 48 hours with 0 hours or no hydro-priming as control.

The seeds were primed with 100 milliliters (ml) of tap water at room temperature (25+-2˚C) in a plastic cup separately for each treatment after measuring their weight. After priming, the seed weight was again measured after removing superficial water to detect water uptake (imbibition) percentage as

$$Water\ imbibition\ \%\ =\ \frac{Weight\ of\ seed\ after\ priming - Weight\ of\ seed\ before\ priming}{Weight\ of\ seed\ before\ priming} \times 100\%$$

Then the soaked seeds were shed dried for 4 hours and sown in the prepared field. The study was conducted in the area of 7.45 m × 7.05 m consisting of 28 plots each of size 0.70

m × 0.35 m. 50 seeds were sown in each treatment plot (in 7 rows each with 7 plants and 1 adjusted manually) with spacing 10 cm × 5 cm. The consecutive treatments were distanced at 1 m apart from each other. The germination percentage was calculated based on daily observation of germinated seeds up to 14 days of sowing.

$$\left(\text{Germination} \% = \frac{\text{Number of germinated seeds}}{\text{Number of the total seed}} \times 100\%\right)$$

After 14 days of sowing, seedling height was measured and respective seedling vigour index (SVI) I and II were calculated as defined by [31].

$$\text{Seedling Vigour Index I(SVI-I)} = \% \text{ germination} \times \text{Seedling Length(cm)}$$

$$\text{Seedling Vigour Index II(SVI-II)} = \% \text{ germination} \times \text{Seedling dry weight(g)}$$

Similarly, seedling height was measured at the interval of four days up to 30 DAS from seven sample plants, each from every row of treatment plots.

## Data analysis

All reported results were manually clustered for treatment-wise under four replications focused on different observable criteria and analyzed using one-way analysis of variance (ANOVA) for randomized complete block design (RCBD) with open-source R software. Duncan's multiple range test at a 5% level of probability was done to compare the means.

## Results

### Water imbibition (%)

The obtained result showed significant variations in seed quality and seedling parameters by bitter gourd seed hydro-priming.

The water imbibition increased with increasing hydro-priming durations. There was a significant difference in increasing hydro-priming durations initially but no significant difference was found after 36 hours of hydro-priming as shown in Table 1.

**Table 1. Effect of seed hydro-priming durations on water imbibition (%) and germination percentage of bitter gourd (*Momordica charantia*).**

| Treatments | Water imbibition (%) by seeds | Germination (%) of seeds at different days | | | | |
|---|---|---|---|---|---|---|
| | | 10 DAS | 11 DAS | 12 DAS | 13 DAS | 14 DAS |
| Control (No hydro-priming) | 0.00[f] | 13[c] | 21[b] | 41.5[b] | 55[b] | 68.00[c] |
| 6 hours of seed hydro-priming | 19.59[e] | 15.5[bc] | 23.5[b] | 43[b] | 55.5[b] | 77.00[b] |
| 12 hours of seed hydro-priming | 31.91[d] | 17[bc] | 28[b] | 45[b] | 66.5[ab] | 80.50[ab] |
| 18 hours of seed hydro-priming | 38.57[c] | 18.5[bc] | 30.5[b] | 45.5[b] | 66.5[ab] | 80.00[ab] |
| 24 hours of seed hydro-priming | 41.88[b] | 19[bc] | 35.5[ab] | 54[ab] | 69[a] | 81.50[ab] |
| 36 hours of seed hydro-priming | 44.83[a] | 20.5[b] | 52[a] | 62[a] | 72[a] | 83.00[ab] |
| 48 hours of seed hydro-priming | 46.34[a] | 26.5[a] | 53[a] | 65.5[a] | 78[a] | 86.50[a] |
| LSD (= 0.05) | 1.82[***] | 6.00[**] | 18.13[**] | 14.18[**] | 11.03[**] | 7.24[**] |
| SEm (±) | 0.23 | 0.763 | 2.307 | 1.804 | 1.403 | 0.921 |
| CV, % | 3.84 | 21.75 | 35.09 | 18.75 | 11.24 | 6.12 |
| Grand mean | 31.87 | 16.93 | 34.43 | 51.57 | 65.29 | 79.5 |

(DAS = days after seeding; LSD (= 0.05) = Least significant difference at 5% probability level; CV = Coefficient of variation; SEm = Standard error of mean. The common letter(s) within the column indicate a non-significant difference based on the Duncan Multiple Range Test (DMRT) at 0.05 level of significance.)

The water imbibition percentage increased simultaneously with an increase in hydro-priming durations. The highest water imbibition percentage was observed in 48 hours of hydro-primed seeds (46.34%) which were at par with 36 hours of hydro-priming (44.83%) but significantly higher than that of seeds hydro-primed for lower durations (41.88%, 38.57%, 31.9%, and 19.59% respectively for 24, 18, 12 and 6 hours hydro-primed seeds.).

## Germination of seeds

The recorded data showed that the germination of bitter gourd seed increased with the increase in the duration of pre-sowing seed hydro-priming as shown in Fig 3. At 10 DAS, the highest germination was recorded on 48 hours of hydro-priming (26.5%), and the lowest being recorded in control (13%). The germination of seeds hydro-primed for 48 hours was significantly higher than other lower hydro-priming durations but the germination of seeds primed for 36 hours (20.5%), 24 hours (19%), 18 hours (18.5%), 12 hours (17%), and 6 hours (15.5%) were statistically similar. Similarly, the germination percentage at 14 days of seeding, the higher germination was observed in hydro-primed seeds than in control. The highest germination was recorded in 48 hours of hydro-primed seeds (86.50%) and the control treatment (68.00%) had the lowest germination percentage followed by seeds primed for 6 hours (77.00). The germination percentage of seeds hydro primed for 12 hours (80.50%), 18 hours (80.00%), 24 hours (81.50%), and 36 hours (83.00%) were statistically at par with 48 hours hydro-pried seeds.

## Height of seedlings

The seedling height was significantly affected by pre-sowing seed hydro-priming durations as observed in Fig 4. With increased hydro-priming durations, there was an increase in seedling height. The tallest seedlings were recorded on 48 hours of seed-hydro-priming, which was significantly higher than other treatments but at par with 24 hours of seed-hydro-priming. At 14 DAS (Days after seeding), observed plant height was found to be significantly higher (5.16 cm) for the seeds hydro-primed for 48 hours than other durations of hydro- priming. The minimum height was recorded in control (3.16 cm) followed by hydro-primed seeds for 6 hours (3.44 cm), 12 hours (3.57 cm), 18 hours (3.82 cm), 24 hours (4.36 cm) and 36 hours (4.64 cm) of hydro-priming Seedling height at 18 hours hydro-priming was at par with 12 hours and 6 hours hydro-priming and control and 6 hours hydro-priming were also statistically at par.

The final height of the seedling was recorded on 30 DAS and it showed a significant difference in height of seedlings. The tallest seedlings were recorded on 48 hours of hydro-priming (18.81 cm) which was significantly higher than other treatments. The shortest seedling height was recorded on control (10.95 cm) followed by seeds hydro-primed for 6 hours (11.79 cm), 12 hours (13.32 cm), 18 hours (16.16 cm), 24 hours (17.02 cm), and 36 hours (18.32 cm). Similar results were obtained at 22 DAS and 26 DAS. The table below (Table 2) showed there was no significant difference among 48 hours and 36 hours and 24 hours of seed hydro-priming at all days of data record.

## Seedling vigour index (SVI)

The result revealed that increased hydro-soaking duration significantly increases the seedling vigour index -I and–II, Fig 5. Seedlings from seeds that were hydro-primed for 48 hours (290.40) were found to be more vigorous in terms of seedling vigour index I than that from other lower hydro-priming durations but at par with 36 hours of hydro-priming (365.38). The lowest seedling vigour Index was obtained in control (209.87) followed by 6 hours (265.34), 12 hours (292.98), 18 hours (305.03), and 24 hours (351.06) hydro-priming.

**Table 2. Effect of seed hydro-priming durations on seedling height and seedling vigour index of bitter gourd (*Momordica charantia*).**

| Treatments | Height (cm) of seedlings at different days | | | | | Seedling Vigour Index (SVI) | |
|---|---|---|---|---|---|---|---|
| | 14 DAS | 18 DAS | 22 DAS | 26 DAS | 30 DAS | SVI-I | SVI-II |
| Control (No hydro-priming) | 3.16$^d$ | 3.73$^d$ | 4.79$^d$ | 7.12$^c$ | 10.95$^d$ | 209.87$^e$ | 29.49$^e$ |
| 6 hours of seed hydro-priming | 3.44$^{cd}$ | 4.18$^c$ | 5.02$^{cd}$ | 7.38$^c$ | 11.79$^{cd}$ | 265.34$^d$ | 39.68$^e$ |
| 12 hours of seed hydro-priming | 3.57$^{cd}$ | 4.40$^c$ | 5.53$^{bc}$ | 8.42$^b$ | 13.32$^c$ | 292.98$^{cd}$ | 43.22$^d$ |
| 18 hours of seed hydro-priming | 3.82$^c$ | 4.94$^b$ | 5.73$^b$ | 8.71$^b$ | 16.16$^b$ | 305.03$^c$ | 45.98$^c$ |
| 24 hours of seed hydro-priming | 4.36$^b$ | 5.34$^a$ | 7.44$^a$ | 10.57$^a$ | 17.02$^{ab}$ | 351.06$^b$ | 47.42$^b$ |
| 36 hours of seed hydro-priming | 4.64$^b$ | 5.48$^a$ | 7.58$^a$ | 10.82$^a$ | 18.32$^{ab}$ | 365.38$^{ab}$ | 57.44$^{ab}$ |
| 48 hours of seed hydro-priming | 5.16$^a$ | 5.64$^a$ | 7.60$^a$ | 11.11$^a$ | 18.81$^a$ | 390.40$^a$ | 62.76$^a$ |
| LSD (= 0.05) | 1.328*** | 0.33*** | 0.64*** | 0.83*** | 2.09*** | 33.33*** | 5.083*** |
| SEm (±) | 0.053 | 0.04 | 0.08 | 0.11 | 0.266 | 4.24 | 0.647 |
| CV, % | 22.218 | 4.65 | 6.93 | 6.1 | 9.26 | 7.2 | 7.59 |
| Grand mean | 4.02 | 4.81 | 6.24 | 9.16 | 15.2 | 311.44 | 46.57 |

(DAS = days after seeding; LSD (= 0.05) = Least significant difference at 5% probability level; CV = Coefficient of variation; SEm = Standard error of mean. The common letter(s) within the column indicate a non-significant difference based on the Duncan Multiple Range Test (DMRT) at 0.05 level of significance.)

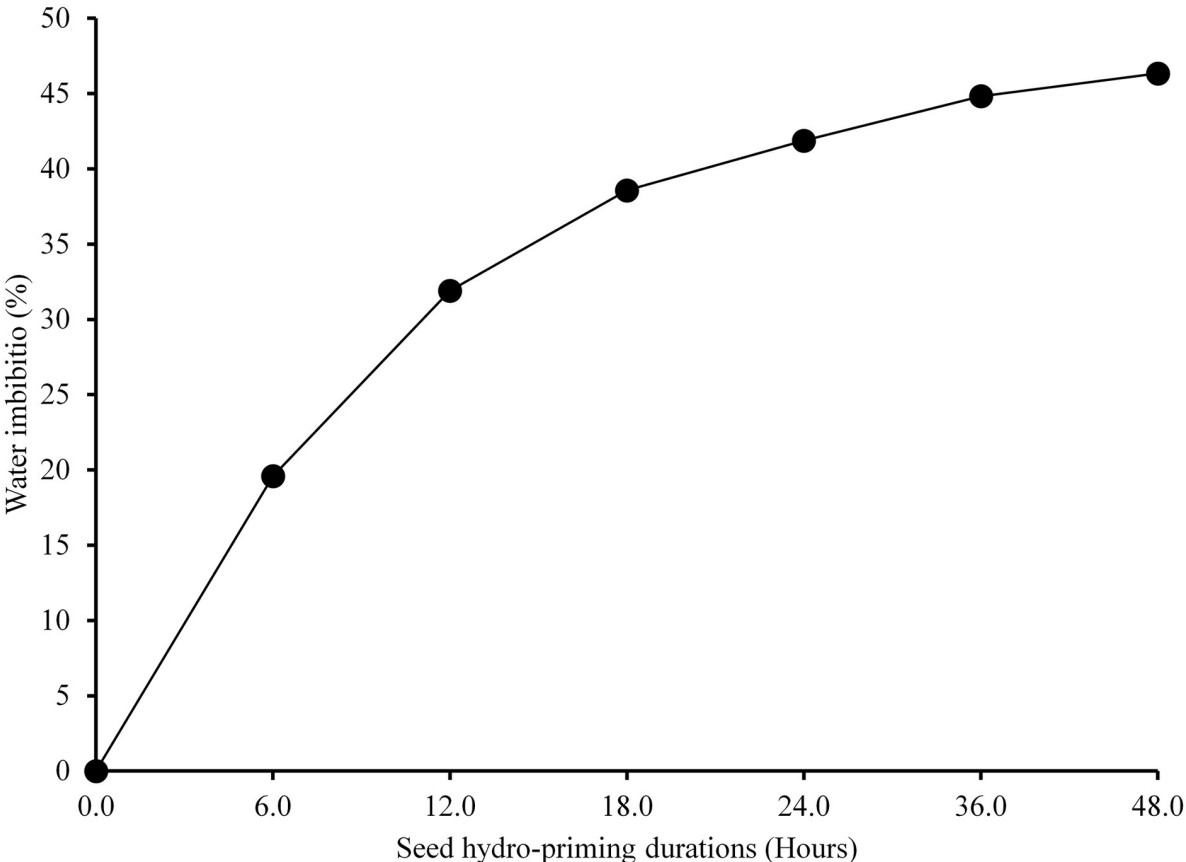

**Fig 2. Water imbibition by bitter gourd seeds at different seed hydro-priming durations.**

Similarly, seedling vigour index -II was found in order of hydro-primed seeds for 48 hours (62.76) followed by 36 hours (57.44), 24 hours 18 hours (45.98), 12 hours (43.22), 6 hours (39.68), and control (29.49), 48 hours hydro-priming and 36 hours hydro-priming, and control and 6 hours hydro-priming being at par.

## Discussion

Initially, seeds have a low moisture content because of which they are inactive. Even so, pre-sowing hydro-priming enhances water imbibition leading to enzyme activation, translocation, and utilization of reserved food materials: accordingly, the findings revealed that the germination and seedling growth is enhanced by hydro-priming (Table 1).

From the water imbibition curve Fig 2, the water imbibition (%) of bitter gourd can be divided into two phases; phase -I and phase-II as described by [32]. Phase-I of the germination had rapid water uptake and it lasted for 24 hours followed by a plateau phase (Phase-II) with little change in water content from 44.83% to 46.34% in 24 hours i.e. after 24 hours to 48 hours of seed priming. During phase I, the water consumed readily hydrates cells and their elements. As a stable state has been achieved between the amount of water consumed by the atmosphere and the amount of water lost by evaporation, there is very little improvement in water absorption in phase II. After the completion of phase II, there is an expansion of the embryo and completes the germination process.

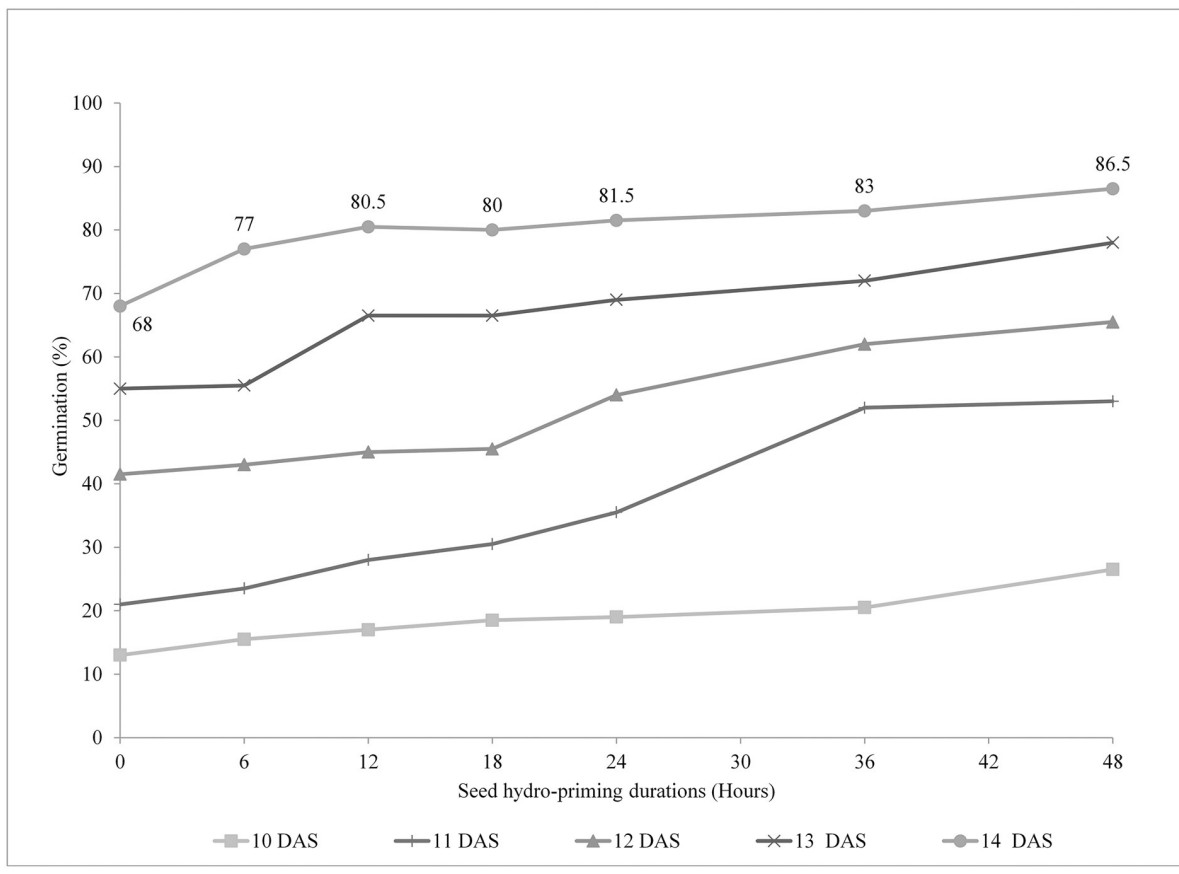

**Fig 3. Germination of bitter gourd seeds at different seed hydro-priming durations.**

The bitter gourd seed coat is very hard and hence took more time to soften the seed coat for germination that results in delayed emergence [3]. Pre-sowing hydro-priming of seed facilitates in softening of seed coat and facilitates the biological process required for germination thereby resulting in early germination [33]. Additionally, it also attenuates initial imbibition variations between the plants, resulting in more uniform germination [34]. So, the germination percentage increases as it also do the time of soaking in water (Fig 3). The result is in parallel with those obtained on different crops [35–37]. The resulting improvement in germination could be due to stimulation of important biochemical adjustments to render seeds ready for radicular protrusion: breakdown of dormancy, hydrolysis, growth inhibitor metabolism, embedding, and enzyme activation [38]. In terms of metabolic and enzymatic repair and recovery of cellular membranes in aged seeds during imbibition, the benefit of bitter gourd seed soaking in tap water is provided, which enables osmotic changes to increase germination. Besides, the thick seed coat is conducive for softening, thereby raising the mechanical restriction influencing embryo development [39, 40].

Early germination and emergence of seeds lead to the rapid growth of seedlings making seedlings taller. Plants produced from primed seeds grow faster than those produced from non-primed plants leading to more height of seedlings (Fig 4). Similarly, [36, 37, 41] also reported better shoot length of cucurbits when they were sown after priming than that of without priming. The beneficial effect of priming on plant growth can be attributed to better root development and consequently the increased nutrient usage capacity that allows for a higher

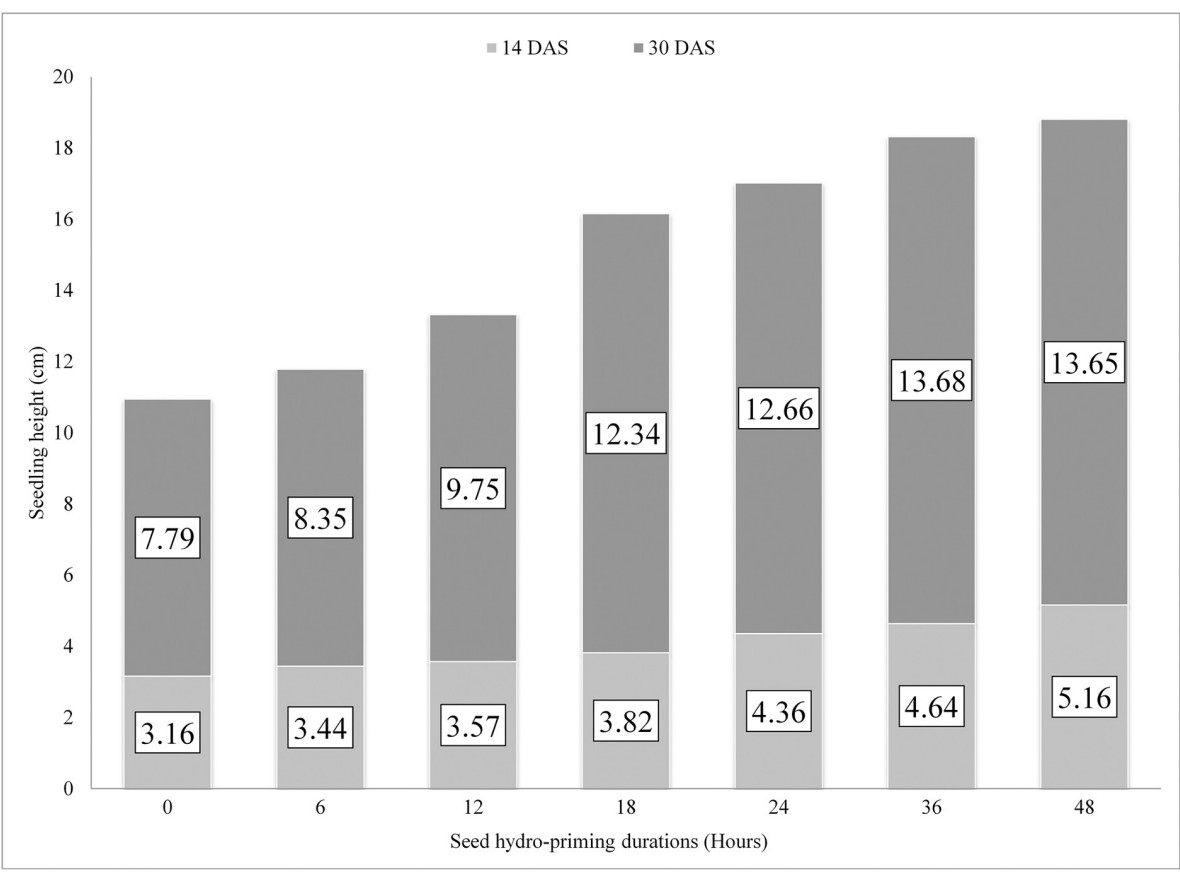

**Fig 4. Seedling height of bitter gourd after 30 DAS at different seed hydro-priming durations.**

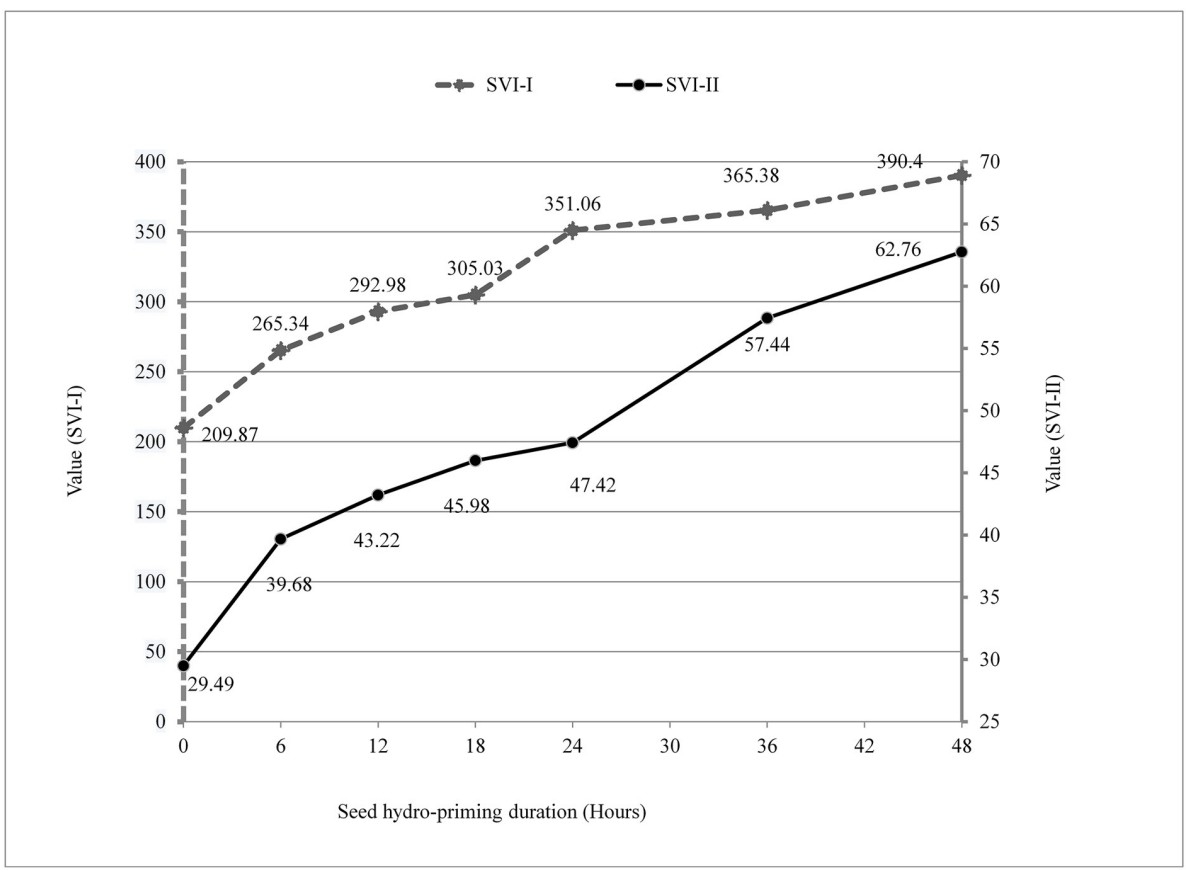

**Fig 5. Seed vigour index (SVI) of bitter gourd at different seed hydro-priming durations.**

relative growth rate [25] and better plant water status regulation [24]. Higher growth of seedlings from primed seed can also be considered with the direct effect of pretreatment on the control of the cell cycle and the mechanisms of cell elongation [42, 43].

Increased hydro-priming durations are attributed to increased seedling vigour index I and II (Fig 5). The seedling vigor index is growing due to reduced imbibition lag time for priming treatments [44]. Similar results were reported during the study of germination and field performance of vegetables; faba bean [45], bitter gourd [27], cucumber [37], and food crops [30, 46]. Priming often induces physiological and biochemical seed modifications during seed treatments [47] and increased amylase production enhances metabolic tasks, resulting in higher seed vigor [46]. Hydro-priming results in the better establishment of the plant root system which is indirectly related to fast growth and improved nutrient absorption that leads to taller plants and higher accumulation of dry matter [45].

## Conclusion

The result of this study revealed that seed hydro-priming durations have a significant impact on germination and seedling growth of bitter gourd. Hydro-priming for 36 hours and 48 hours significantly enhances water imbibition thereby causing higher germination, seedling height, and seedling vigour index, both of which are statistically similar. The research widened the possibility of recommending 36-hours hydro-priming of seeds before sowing as a regular

practice to farmers to secure higher seed germination and better growth and development of seedling through organic, cost-effective, easily available, and less tedious work.

Hence seed hydro-priming can be used as a simple and cost-effective procedure to improve seed germination and seedling quality under agro-climatic conditions of Surkhet district, Karnali Province, Nepal. Still, additional studies are required to confirm a similar effect on the large, commercial field.

## Supporting information

**S1 Table. ANOVA of water uptake (%) in bitter gourd seeds influenced by hydro-priming durations in Birendranagar, Surkhet, 2020.**
(TIF)

**S2 Table. ANOVA of germination percentage of bitter gourd at different days after seeding (DAS) under open field condition in Birendranagar, Surkhet, 2020.**
(TIF)

**S3 Table. ANOVA on seedling height of bitter gourd at different DAS under open field condition in Birendranagar, Surkhet, 2020.**
(TIF)

**S4 Table. ANOVA on seed vigour index (SVI) of bitter gourd at 14 DAS under open field condition in Birendranagar, Surkhet, 2020.**
(TIF)

## Author Contributions

**Conceptualization:** Binod Adhikari, Pankaj Raj Dhital, Sambat Ranabhat, Hari Poudel.

**Data curation:** Binod Adhikari.

**Formal analysis:** Binod Adhikari, Pankaj Raj Dhital.

**Investigation:** Binod Adhikari.

**Methodology:** Binod Adhikari, Pankaj Raj Dhital, Sambat Ranabhat, Hari Poudel.

**Project administration:** Binod Adhikari.

**Resources:** Binod Adhikari.

**Software:** Binod Adhikari.

**Supervision:** Binod Adhikari, Pankaj Raj Dhital.

**Validation:** Binod Adhikari, Sambat Ranabhat, Hari Poudel.

**Visualization:** Binod Adhikari, Pankaj Raj Dhital.

**Writing – original draft:** Binod Adhikari, Pankaj Raj Dhital, Sambat Ranabhat, Hari Poudel.

**Writing – review & editing:** Binod Adhikari, Pankaj Raj Dhital, Hari Poudel.

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
