## [Decision Letter · Decision Letter 0]

6 May 2021

PONE-D-21-08775

Effect of seed hydro-priming durations on germination and seedling growth of bitter gourd (Momordica charantia)

PLOS ONE

Dear Dr. Adhikari,

Thank you for submitting your manuscript to PLOS ONE. After careful consideration, we feel that it has merit but does not fully meet PLOS ONE’s publication criteria as it currently stands. Therefore, we invite you to submit a revised version of the manuscript that addresses the points raised during the review process.

Once the manuscript has been revised and resubmitted, it will be sent to the reviewers for additional comments and to ensure that all of their concerns have been addressed. A common issue that both reviewers noted was the lack of a discussion of current work, as evidenced by a lack of citation of more recent papers. This is a problem that definitely needs to be addressed. The other issues that reviewers mention are also important and I strongly recommend that you make the necessary corrections according to their suggestions in order for your manuscript to be considered for publication.

We look forward to receiving your revised manuscript.

Kind regards,

Craig Eliot Coleman, PhD

Academic Editor

PLOS ONE

Journal Requirements:

3. Please include a copy of Table 7 which you refer to in your text (line130).

Reviewers' comments:

Reviewer's Responses to Questions

**Comments to the Author**

1. Is the manuscript technically sound, and do the data support the conclusions?

Reviewer #1: Yes

Reviewer #2: Partly

2. Has the statistical analysis been performed appropriately and rigorously? 

Reviewer #1: Yes

Reviewer #2: No

3. Have the authors made all data underlying the findings in their manuscript fully available?

Reviewer #1: Yes

Reviewer #2: Yes

4. Is the manuscript presented in an intelligible fashion and written in standard English?

Reviewer #1: No

Reviewer #2: No

5. Review Comments to the Author

Reviewer #1: Comments

The authors found in the experiment, seed hydro-priming was found to be effective for increasing germination and seedling growth in bitter gourd only on the basis of some morphological parameters. However, there is a space to add more morphological parameters to conclude the results. For publication in such prestigious journal, I recommend major revisions.

Line 21. In abstract Please mention all the priming durations otherwise abstract will not be self-explanatory. Before going to results its need to specify the methods adopted in the study.

Line 36. The crop is of high economic and nutritional importance. Please rewrite the sentence.

The introduction is poorly citied. For instance, in case of hydro priming only one citation is written. However, recently many work has been conducting on priming and hydro-priming. Please cite more recent reference and I think also space for writing some introductory notes on priming before hydro-priming.

Authors should more careful on grammatical mistakes in the introduction.

Line 91-92 Authors mentioned no significant difference was found after 36 hours of hydro-priming as shown in Fig 1. But in figure its looks not clear because there is no sign of statistical analysis.

The experiments lack of some data such as minerals homeostasis uptake data, water related traits, can be checked because author discussed priming can be attributed to an increased nutrient usage capacity that allows for a higher relative growth rate and better plant water status regulation. Also, many others morphological parameters can be added.

In the discussion, results did not discusses properly authors only focus the literature. Discussion section should be more clarified with discussions and results.

Overall, the writing lack of many references, more literature should be added. On the other hand, climatic condition in the experiment time was an important factor. However, there is no information about climatic condition of the experimental field. Moreover, one trial in the field for one year is not sufficient to conclude, it needs at least 2-3 trial on 2-3 consecutive years.

Reviewer #2: 1. Give reference for the statement given on line number 43-44.

2. Please provide experimental site details i.e. location, soil, weather etc at the beginning of materials and methods.

3. Mention source of the seed, its initial germination percentage and class of the seed at line number 67.

4. The formula given on line number 71 seems wrong. It should be weight of seed after priming - weight of seed before priming/weight of seed before priming.

5. A reference of Table 7 has been given on line number 130 however there is no Table 7 in the whole text.

6. Give reference for this statement given on line 172-174.

7. Why don’t you recommend 36 hours seed priming instead of 48 hours which has similar results in most cases?

8. Most of the references are old which can be replaced by latest ones.

6. PLOS authors have the option to publish the peer review history of their article (what does this mean?). If published, this will include your full peer review and any attached files.

Reviewer #1: **Yes: **Mohammad Saidur Rhaman

Reviewer #2: **Yes: **Prof. Dr. Muhammad Arif

---

## [Author Response · Author response to Decision Letter 0]

22 Jun 2021

Reviewer #1: Comments

• Line 21. In abstract Please mention all the priming durations otherwise abstract will not be self-explanatory. Before going to results its need to specify the methods adopted in the study.

Thank you for pointing this out. We have included the priming durations.

• Line 36. The crop is of high economic and nutritional importance. Please rewrite the sentence.

As suggested by reviewer, we have revised the sentence. The revised sentence reads as “The crop is of high nutritional and medicinal importance. Its immature fruit is rich source of dietary fibers, minerals and Vitamins (C and A) which also acts as a blood purifier and is highly beneficial to diabetes patients. Likewise, it also has anti-carcinogenic property and can be used against multiple cancer forms as a cytostatic agent”. 

• The introduction is poorly citied. For instance, in case of hydro priming only one citation is written. However, recently many work has been conducting on priming and hydro-priming. Please cite more recent reference and I think also space for writing some introductory notes on priming before hydro-priming.

We have added the suggested content to the manuscript (line 49-62).

• Authors should more careful on grammatical mistakes in the introduction.

Thank you for your suggestions. We have worked out on this and corrected the errors.

• Line 91-92 Authors mentioned no significant difference was found after 36 hours of hydro-priming as shown in Fig 1. But in figure its looks not clear because there is no sign of statistical analysis.

Thank you for pointing this out. We have added the respective table with necessary sign of statistical analysis (Table 1 and 2).

• In the discussion, results did not discusses properly authors only focus the literature. Discussion section should be more clarified with discussions and results.

A suggested by the reviewer, we have clarified discussion section with discussions and results.

• The experiments lack of some data such as minerals homeostasis uptake data, water related traits, can be checked because author discussed priming can be attributed to an increased nutrient usage capacity that allows for a higher relative growth rate and better plant water status regulation. Also, many others morphological parameters can be added.

Thank you for pointing this out. We have taken data related to root length and seedling dry weight too along with water imbibition, germination percentage and seedling height. Since, they are the major factor in calculating seedling vigour index I and II, we haven’t included them separately.

• Overall, the writing lack of many references, more literature should be added.

As per suggestions, we have read more literatures and included in the manuscript.

• On the other hand, climatic condition in the experiment time was an important factor. However, there is no information about climatic condition of the experimental field.

We think this is the vital suggestion. We have included climatic details in the line 79-87.

• Moreover, one trial in the field for one year is not sufficient to conclude, it needs at least 2-3 trial on 2-3 consecutive years.

Reviewer #2: 

• Give reference for the statement given on line number 43-44.

• Give reference for this statement given on line 172-174.

Thank you for pointing this out. We have added the references for respective sentences in given line numbers.

• Please provide experimental site details i.e. location, soil, weather etc at the beginning of materials and methods.

We think this is an important suggestion. We have included those details on line number 79-87.

• Mention source of the seed, its initial germination percentage and class of the seed at line number 67.

As per suggestions we have included the detail on line number 90 and 91.

• The formula given on line number 71 seems wrong. It should be weight of seed after priming - weight of seed before priming/weight of seed before priming.

The formula is corrected as per the reviewer’s suggestion.

• A reference of Table 7 has been given on line number 130 however there is no Table 7 in the whole text.

Thank you for pointing this out. We have now added the tables (Table 1 and 2) to for clarification of the reference.

• Why don’t you recommend 36 hours seed priming instead of 48 hours which has similar results in most cases?

We agree with the reviewer’s suggestion. Accordingly, we have revised our conclusion to recommend 36 hours of hydro-priming for bitter gourd seed prior to seeding/sowing.

• Most of the references are old which can be replaced by latest ones.

As per the suggestions, we have cited more recent literatures.

---

## [Decision Letter · Decision Letter 1]

13 Jul 2021

Effect of seed hydro-priming durations on germination and seedling growth of bitter gourd (Momordica charantia)

PONE-D-21-08775R1

Dear Dr. Adhikari,

We’re pleased to inform you that your manuscript has been judged scientifically suitable for publication and will be formally accepted for publication once it meets all outstanding technical requirements.

Kind regards,

Craig Eliot Coleman, PhD

Academic Editor

PLOS ONE

Additional Editor Comments (optional):

Reviewers' comments:

Reviewer's Responses to Questions

**Comments to the Author**

1. If the authors have adequately addressed your comments raised in a previous round of review and you feel that this manuscript is now acceptable for publication, you may indicate that here to bypass the “Comments to the Author” section, enter your conflict of interest statement in the “Confidential to Editor” section, and submit your "Accept" recommendation.

Reviewer #1: (No Response)

2. Is the manuscript technically sound, and do the data support the conclusions?

Reviewer #1: Yes

3. Has the statistical analysis been performed appropriately and rigorously? 

Reviewer #1: Yes

4. Have the authors made all data underlying the findings in their manuscript fully available?

Reviewer #1: Yes

5. Is the manuscript presented in an intelligible fashion and written in standard English?

Reviewer #1: Yes

6. Review Comments to the Author

Reviewer #1: (No Response)

7. PLOS authors have the option to publish the peer review history of their article (what does this mean?). If published, this will include your full peer review and any attached files.

Reviewer #1: **Yes: **Mohammad Saidur Rhaman

---

## [Editor Report · Acceptance letter]

19 Jul 2021

PONE-D-21-08775R1 

Effect of seed hydro-priming durations on germination and seedling growth of bitter gourd (*Momordica charantia*) 

Dear Dr. Adhikari:

I'm pleased to inform you that your manuscript has been deemed suitable for publication in PLOS ONE. Congratulations! Your manuscript is now with our production department. 

Kind regards, 

on behalf of

Dr. Craig Eliot Coleman 

Academic Editor

PLOS ONE